# Identification of type 2 diabetes- and obesity-associated human β-cells using deep transfer learning

**Gitanjali Roy[1], Rameesha Syed[1†], Olivia Lazaro[1†], Sylvia Robertson[1], Sean D McCabe[2], Daniela Rodriguez[1], Alex M Mawla[3], Travis S Johnson[1,2]\*, Michael A Kalwat[1,4]\***

[1]Indiana Biosciences Research Institute, Indianapolis, United States; [2]Department of Biostatistics and Health Data Science, Indiana University School of Medicine, Indianapolis, United States; [3]Department of Neurobiology, Physiology and Behavior, College of Biological Sciences, University of California, Davis, Davis, United States; [4]Center for Diabetes and Metabolic Diseases, Indiana University School of Medicine, Indianapolis, United States

\*For correspondence:
johnstrs@iu.edu (TSJ);
mkalwat@indianabiosciences.
org (MAK)

†These authors contributed
equally to this work

Competing interest: The authors
declare that no competing
interests exist.

Reviewing Editor: Lori Sussel,
University of Colorado Anschutz
Medical Campus, United States

## eLife assessment

This is a **useful** study that applies deep transfer learning to assign patient-level disease attributes to single cells of T2D and non-diabetic patients, including obese patients. This analysis identified a single cluster of T2D-associated β-cells; and two subpopulations of obese- β-cells derived from either non-diabetic or T2D donors. The findings were validated at the protein level using immuno-histochemistry on islets derived from non-diabetic and T2D organ donors, contributing **solid** experimental evidence for the computational analyses.

**Abstract** Diabetes affects >10% of adults worldwide and is caused by impaired production or response to insulin, resulting in chronic hyperglycemia. Pancreatic islet β-cells are the sole source of endogenous insulin, and our understanding of β-cell dysfunction and death in type 2 diabetes (T2D) is incomplete. Single-cell RNA-seq data supports heterogeneity as an important factor in β-cell function and survival. However, it is difficult to identify which β-cell phenotypes are critical for T2D etiology and progression. Our goal was to prioritize specific disease-related β-cell subpopulations to better understand T2D pathogenesis and identify relevant genes for targeted therapeutics. To address this, we applied a deep transfer learning tool, DEGAS, which maps disease associations onto single-cell RNA-seq data from bulk expression data. Independent runs of DEGAS using T2D or obesity status identified distinct β-cell subpopulations. A singular cluster of T2D-associated β-cells was identified; however, β-cells with high obese-DEGAS scores contained two subpopulations derived largely from either non-diabetic (ND) or T2D donors. The obesity-associated ND cells were enriched for translation and unfolded protein response genes compared to T2D cells. We selected CDKN1C and DLK1 for validation by immunostaining in human pancreas sections from healthy and T2D donors. Both CDKN1C and DLK1 were heterogeneously expressed among β-cells. CDKN1C was increased in β-cells from T2D donors, in agreement with the DEGAS predictions, while DLK1 appeared depleted from T2D islets of some donors. In conclusion, DEGAS has the potential to advance our holistic understanding of the β-cell transcriptomic phenotypes, including features that distinguish β-cells in obese ND or lean T2D states. Future work will expand this approach to additional human islet omics datasets to reveal the complex multicellular interactions driving T2D.

## Introduction

The advent of high-throughput single-cell RNA sequencing (scRNA-seq) has enabled the generation of an array of single-cell atlases from pancreatic islets. These findings have expanded our understanding of the major cell types of the pancreas along with how they are implicated in both type 1 diabetes (T1D) and type 2 diabetes (T2D). Notably, these studies have (1) identified multiple reliable transcriptomic markers for endocrine and exocrine pancreatic cell types; (2) provided insight into novel subtypes of cells; and (3) generated large cellular atlases spurring innovation in the development of single-cell methods and analysis. scRNA-seq analysis has enabled a more robust characterization of islet cell heterogeneity, which may underlie diversity in diabetes risk and drug response (*Baron et al., 2016*; *Li et al., 2016*; *Wang et al., 2016*; *Lawlor et al., 2017*; *Fang et al., 2019*). In published comparisons between the islet single-cell transcriptomes of humans versus mice and pigs, differences were observed in the relative proportion of major cell types as well as in many cell type-specific genes (*Tritschler et al., 2022*). These findings further support a focus on integrating human islet transcriptomic data to delineate disease processes and identify therapeutic targets and biomarkers.

As bulk RNA-seq and scRNA-seq human islet datasets have become increasingly available, so too has the need for new computational tools to integrate the transcriptomics and donor metadata. Some sets of islet scRNA-seq data have been combined and made searchable through web portals to browse datasets and compare cell types and marker genes (*Tritschler et al., 2022*; *Segerstolpe et al., 2016*; *Mawla and Huising, 2019*; *Elgamal et al., 2023*). To further utilize these types of integrated transcriptomic data and donor metadata, we previously developed DEGAS as a flexible deep transfer learning framework that can be used to overlay disease status, survival hazard, drug response, and other clinical information directly onto single cells. Machine learning tools like DEGAS have been primarily used in cancer datasets, but not in human pancreatic islets until now.

The reasons why some obese individuals succumb to T2D while others do not likely involve both genetic and environmental factors, but the factors that underlie this transition are incompletely understood (*Fabbrini et al., 2015*). Analyses of human genomics and islet transcriptomics indicate many β-cell genes have causal roles in T2D (*Voight et al., 2010*; *Saxena et al., 2007*; *Scott et al., 2007*; *Rottner et al., 2023*; *Kim et al., 2023*). Subsets of β-cells which are more resilient or susceptible to failure under the secretory pressure of insulin resistance may be uncovered by combining transcriptomics with machine learning approaches like DEGAS. Here we have implemented DEGAS to predict T2D- and obesity-associated subclusters of human pancreatic islet β-cells using a combination of publicly available scRNA-seq and bulk RNA-seq human islet data and associated donor metadata. Through this analysis, we sought to identify genes implicated in T2D and obesity that were up- or downregulated in subpopulations of β-cells identified by DEGAS and to validate our findings at the protein level using immunohistochemistry of pancreas tissue from non-diabetic (ND) and T2D organ donors. Our findings applying DEGAS to islet data have implications for β-cell heterogeneity in T2D and obesity.

## Results

### Differentially expressed genes from T2D bulk RNA-seq human islets show inflammatory signature and correlate with independent islet datasets

We selected bulk RNA-seq data from *Marselli et al., 2020* (GSE159984) which contains 58 ND and 27 T2D samples with associated metadata for BMI, age, and sex (*Marselli et al., 2020*; *Figure 1A–C*, *Supplementary file 1*). To determine the suitability of this data for DEGAS, we reanalyzed the read count matrix from GSE159984 using the edgeR likelihood ratio test to identify up- and downregulated genes, visualized in both volcano (*Figure 1D*) and MD plot (*Figure 1E*) formats to highlight fold change versus significance and versus overall expression levels, respectively (*Supplementary file 2*). We observed altered expression of genes well known to be dysregulated in T2D, including *IAPP* (*Chen et al., 2018*; *Folli et al., 2018*), *PAX4* (*Lorenzo et al., 2015*; *Collombat et al., 2003*), *SLC2A2* (*Dupuis et al., 2010*), *FFAR4* (*Wu et al., 2021*), and *ENTPD3* (*Docherty et al., 2021*; *Syed et al., 2013*; *Figure 1D, E*). Gene set enrichment analysis (GSEA) indicated that the gene expression profile of T2D human islets from this dataset was associated with an inflammatory response phenotype (*Figure 1F*). To compare our analysis of Marselli et al. with an independent cohort, we selected a bulk

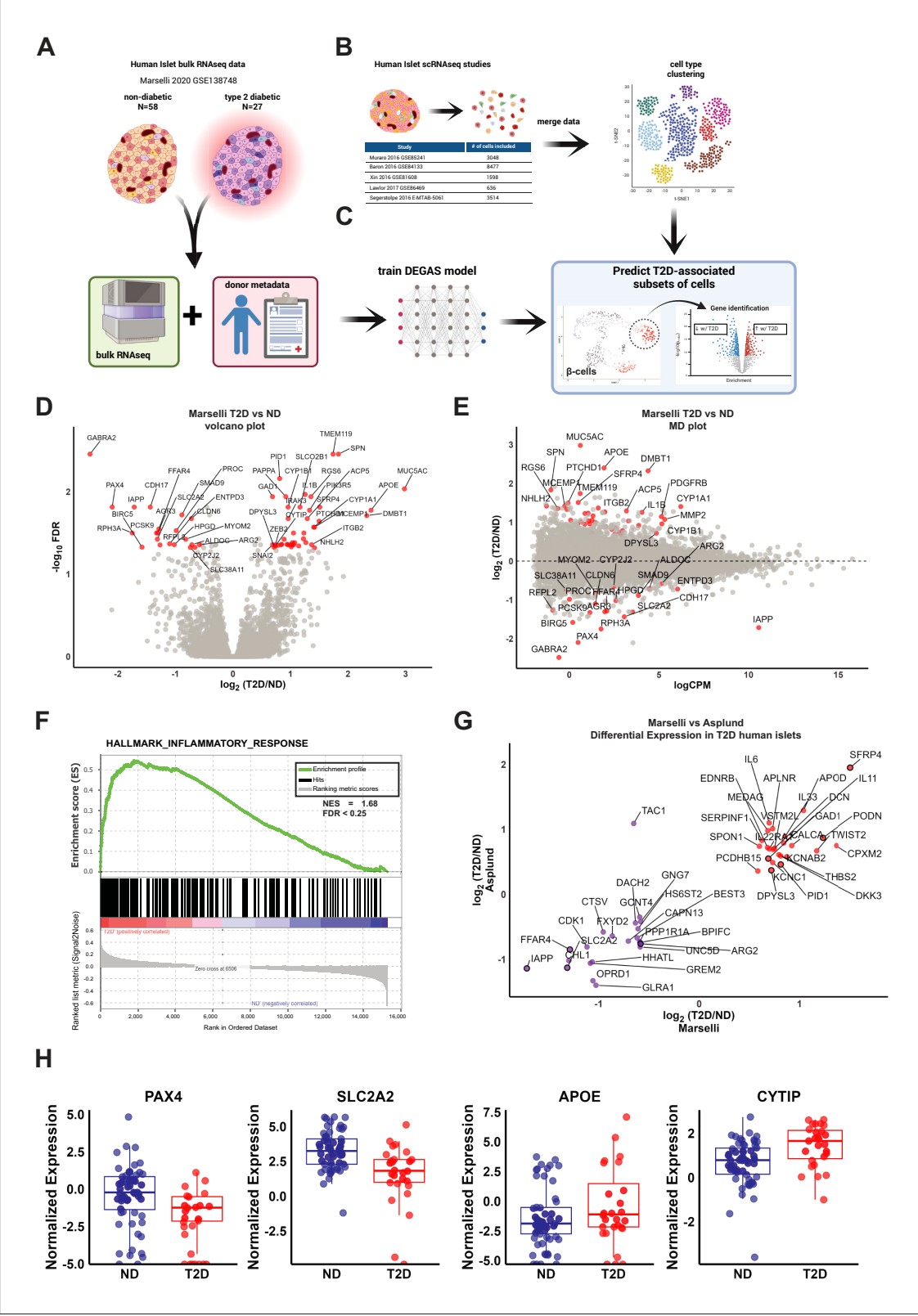

**Figure 1.** Data acquisition and workflow to train DEGAS using human pancreatic islet transcriptomic data for prediction of type 2 diabetes (T2D)-associated cells. (**A**) Read count data for human islet bulk RNA-seq from GSE138748 was downloaded, processed, and matched with donor metadata. The dataset included 58 non-diabetic and 27 T2D samples. (**B**) Human islet single-cell RNA-seq (scRNA-seq) count matrices were generated by realignment of reads and the datasets were integrated in Seurat. (**C**) DEGAS transfers the bulk donor expression data and clinical trait information

*Figure 1 continued on next page*

Figure 1 continued

to individual cells in the single-cell matrix for the purpose of prioritizing cells. This allows cells to be assigned scores which can then be thresholded for downstream analysis. Bulk RNA-seq data from Marselli et al. was analyzed by edgeR and displayed as both (**D**) volcano plot and (**E**) MA plot. Differentially expressed genes (DEGs) are those with p < 0.05 and >1.5 fold changed and are highlighted in red. The MA plot in (**E**) provides a sense of relative transcript abundance among DEGs. (**F**) Gene set enrichment analysis (GSEA) results for Hallmark_Inflammatory_Response show an enrichment for DEGs in T2D human islets. (**G**) A simple comparison of up- and downregulated genes in T2D human islets in RNA-seq data between Marselli et al. and Asplund et al. shows consistent findings. Significant DEGs from Marselli et al. are outlined in black. (**H**) Selected significant genes from (**D**) that were also found on the Type 2 Diabetes Knowledge Portal database of known diabetes effector genes. The data for selected genes across all donors are displayed in box and whiskers plots. Whiskers extend to 1.5 times the interquartile range.

The online version of this article includes the following figure supplement(s) for figure 1:

**Figure supplement 1.** Islet Gene View results of DEGs found in both Marselli et al. and Asplund et al.

RNA-seq human islet transcriptomic analysis from *Asplund et al., 2022*. We compared the published supplementary table of $\log_2$-fold changes for differentially expressed genes to the Marselli dataset (*Figure 1G*; *Supplementary file 2*). Significant differentially expressed genes from both Marselli and Asplund data included upregulation of *SFRP4* and *PODN* (*Figure 1—figure supplement 1A, B*), and downregulation of *UNC5D* and *FFAR4* (*Figure 1—figure supplement 1C, D*). We also found substantial agreement in the directional changes of most other differentially expressed genes from both datasets, indicating that the Marselli islet data represents a suitable cohort for DEGAS analysis. Finally, we compared up- and downregulated genes from T2D islets in the Marselli dataset with T2D effector genes from the T2D Knowledge Portal (*Costanzo et al., 2023*). Downregulated genes *PAX4* and *SLC2A2* and upregulated genes *APOE* each have evidence of a strong or causal role in T2D at the genetic level, while *CYTIP* was upregulated in T2D islets and has intron variant SNPs associated with T2D (e.g. rs13384965) (*Forgetta et al., 2022*; *Figure 1H*).

## DEGAS revealed T2D- and obesity-associated β-cells and marker genes within integrated human islet scRNA-seq data

We obtained five realigned scRNA-seq datasets (*Mawla and Huising, 2019*) including Baron (GSE84133), Muraro (GSE85241), Segerstolpe (E-MTAB-5061), Lawlor (GSE86469), and Xin (GSE81608), and integrated them using Seurat 4.3.0 to subtype each of the major cell types for use with DEGAS. All five datasets were successfully integrated, resulting in 17,273 cells (*Figure 2A, B*). Different cell types were clearly stratified from the scRNA-seq data, as indicated by distinct clusters of cell types enriched for their respective markers including β-cells (*INS*), α-cells (*GCG*), δ-cells (*SST*), ε-cells (*GHRL*), PP-cells (*PPY*), acinar cells (*REG1A*, *PRSS1*), ductal cells (*KRT19*, *CFTR*), stellate cells (*COL6A1*), endothelial cells (*PLVAP*), and mesenchymal cells (*CD44*) (*Figure 2C*). Other parameters including sex, BMI, age, and T2D status were represented in the merged dataset (*Figure 2D–G*). Markers of T2D versus ND for all cells and for only β-cells were also extracted for downstream comparisons (*Figure 2—figure supplement 1A, B*). There was limited overlap between the differentially expressed genes in Marselli et al. compared to T2D versus ND for all single cells, but included *IAPP*, *ENTPD3*, and *FFAR4*.

Next, we implemented DEGAS (*Johnson et al., 2022*), a unique tool that leverages the increased sequencing depth, larger number of clinical covariates, and sample sizes of bulk sequencing data. We used the deep learning architecture within DEGAS to project bulk islet sequencing data into the same latent representation as single-cell islet data via domain adaptation. From this common latent representation, clinical covariates from the bulk sequencing data (e.g. T2D status and BMI) were projected onto single cells in a process called transfer learning. This process generates unitless disease-association scores for each single cell. In parallel, we applied DEGAS to the merged scRNA-seq data using either T2D or obesity as the scoring feature.

### T2D-DEGAS

First, we next mapped the T2D-association scores onto the single cells (*Figure 3A*). β-Cells had a wide distribution of scores, possibly reflecting β-cell heterogeneity or altered β-cell gene expression after onset of T2D (*Figure 3B*; *Weir et al., 2020*). We focused on differential T2D-association scores only among β-cells by subsetting them based on *INS* expression and superimposed the T2D-association scores onto each β-cell in the UMAP plot (*Figure 4A*). β-Cells with higher scores (pink coloration)

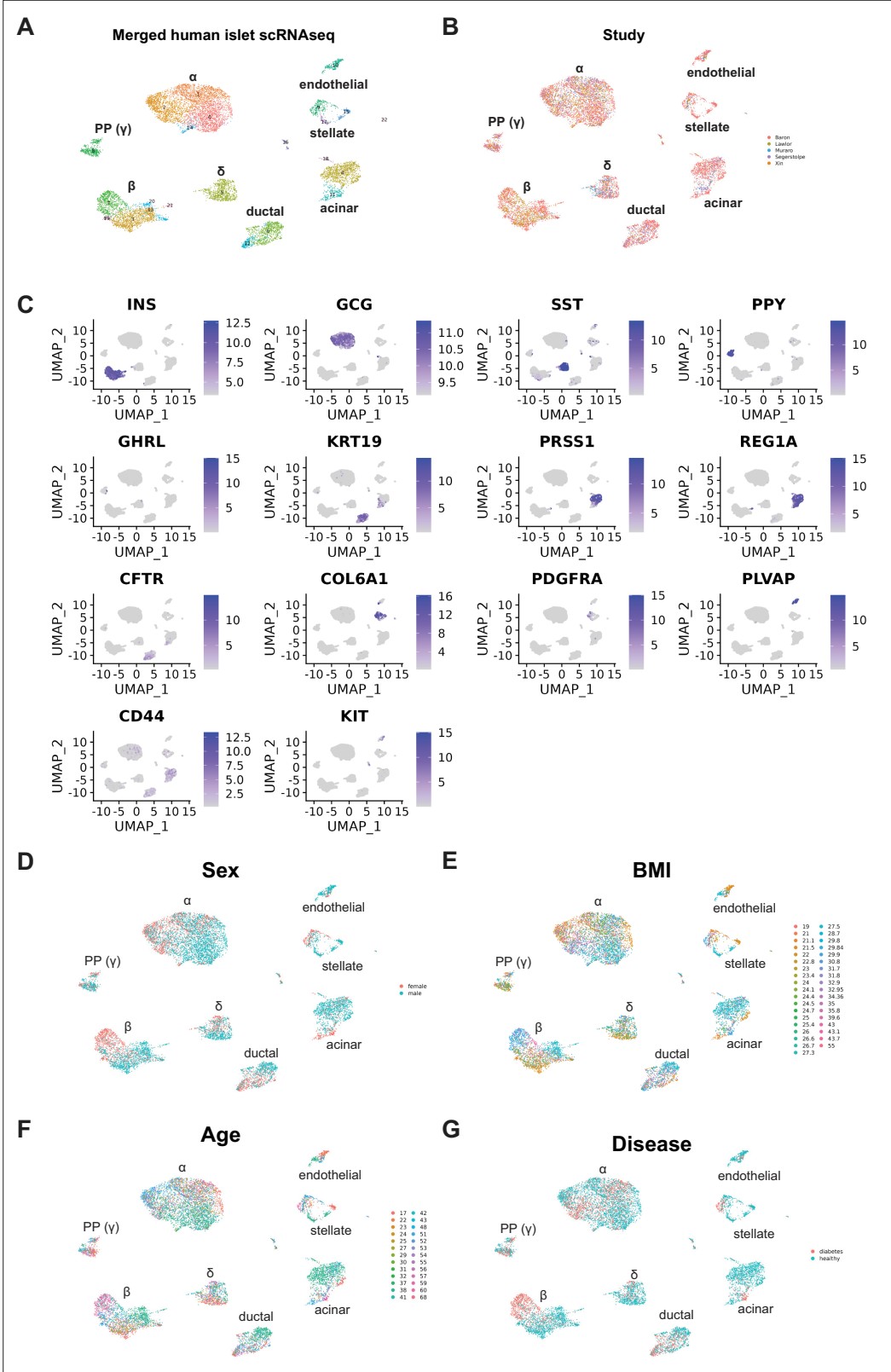

**Figure 2.** Validation of merged scRNA-seq datasets from five human islet studies. (**A**) Seurat clustering of merged human islet scRNA-seq data from *Figure 1B*. (**B**) Study identifiers overlaid on single-cell plot. (**C**) Gene expression plots for the major endocrine and exocrine pancreas genes to demonstrate successful clustering of specific cell types. Metadata was overlaid onto single-cell UMAP plots for sex (**D**), BMI (**E**), age (**F**), and diabetes status (**G**).

*Figure 2 continued*

The online version of this article includes the following figure supplement(s) for figure 2:

**Figure supplement 1.** Human islet single-cell RNA-seq clusters and diabetes marker genes for all cells and β-cells.

transcriptionally associate with human islet T2D expression profiles more than the lower scoring (black) β-cells. Next, we explored median and quantile thresholds for the β-cell T2D-association scores (*Figure 4—figure supplement 1A, B*) and continued our analysis with the cells grouped into high (upper 20%), medium, and low (lower 20%) categories (*Figure 4B*). We compared the high versus low β$^{T2D-DEGAS}$ groups to genes enriched in either subset of β-cells (*Figure 4C*, *Supplementary file 3*). High-scoring β$^{T2D-DEGAS}$ cells had elevated expression of *CDKN1C* (p57$^{Kip2}$) which binds and inhibits G1 cyclin/CDK complexes and is also involved in cellular senescence (*Stampone et al., 2018*; *Ou et al., 2019*), and reduced expression of *IAPP*, consistent with results from bulk T2D islet expression data (*Figure 1G*). These findings show that the DEGAS can successfully identify diabetes-relevant genes enriched in specific subclusters.

As an additional way to highlight genes that were identified by DEGAS, we excluded genes that were differentially expressed between all ND and T2D β-cells (based on the merged scRNA-seq donor metadata; *Figure 4—figure supplement 2D*, *Supplementary file 3*). This process left over 90% of significant differentially expressed genes remaining which were only identified through DEGAS analysis (*Figure 4D*). This emphasized the advantage of using DEGAS to generate continuous variables (disease-association scores) which enable thresholding and prioritizing cellular subpopulations. Gene ontology (GO) analysis on high-scoring β$^{T2D-DEGAS}$ cells showed enrichment in biological process categories including neutrophil degranulation, cell growth, and negative regulation of cell development (*Figure 4E*, *Supplementary file 4*). These categories were driven in part by immune-related genes (e.g. *HLA-B*, *PSAP*) and cell cycle regulator *BTG1*. Conversely, GO analysis on low-scoring β$^{T2D-DEGAS}$ cells showed enrichment of categories primarily related to membrane protein targeting, translation, and mitochondrial electron transport (*Figure 4F*, *Supplementary file 4*). The top six enriched GO categories were driven largely by *RPS/RPL* genes. We also performed ranked-list GSEA on these same high- versus low-scoring β$^{T2D-DEGAS}$ subpopulations, which largely agreed with the GO analysis (*Figure 4G*). We observed significant positive enrichment for hypoxia, TNFα signaling, estrogen response, and glycolysis (*Figure 4—figure supplement 1C*), while significant negative enriched pathways included overlapping genes in oxidative phosphorylation and Myc-target pathways (*Figure 4—figure supplement 1D*). Examples of genes that drive the GO and GSEA categories are shown (*Figure 4H*), and heterogeneity in the expression in a selection of these genes can be seen in β-cell UMAP plots (*Figure 4—figure supplement 1E*).

## Obese-DEGAS

Second, we applied DEGAS using obesity as the disease feature. Obesity and T2D are related, but not mutually inclusive. We surmised that obesity-association scores in β-cells may highlight unique subpopulations compared to the β$^{T2D-DEGAS}$ cells. Therefore, we categorized bulk RNA-seq donors by BMI (lean, <25; overweight, 25–30; obese, >30) and ran the DEGAS analysis based on these categories to calculate BMI-association scores similar to our approach with T2D. Overlaying the BMI-DEGAS scores onto the islet single-cell data showed a distinct pattern of cell labeling between them (*Figure 4—figure supplement 2A, B*). The cell types with the highest obesity-association scores were β-cells, acinar cells, and stellate cells (*Figure 5A, B*).

We next reclustered β-cells with overlaid lean, overweight, and obesity-association scores (*Figure 4—figure supplement 2C, D*). It is notable that the subpopulation of β$^{lean-DEGAS}$ cells appears to overlap with the subpopulation of low-scoring β$^{T2D-DEGAS}$ cells, suggesting that single β-cells share features of low BMI and potentially lower T2D association. β$^{overweight-DEGAS}$ scores did not highlight any subpopulation of β-cells (*Figure 4—figure supplement 2D*). However, we observed two subpopulations of high-scoring β$^{obese-DEGAS}$ cells (*Figure 6A*), a feature that did not occur with β$^{T2D-DEGAS}$ cells (*Figure 4B*). We observed that a distinguishing difference between these two groups of obesity-associated β-cells was the donor diabetes status (ND vs. T2D) from within the single-cell metadata. We applied a median quantile threshold to extract these two subpopulations (*Figure 4—figure supplement 2E*) and classified the cell clusters based on their donor diabetes status (*Figure 6B*, *Figure 4—figure supplement*

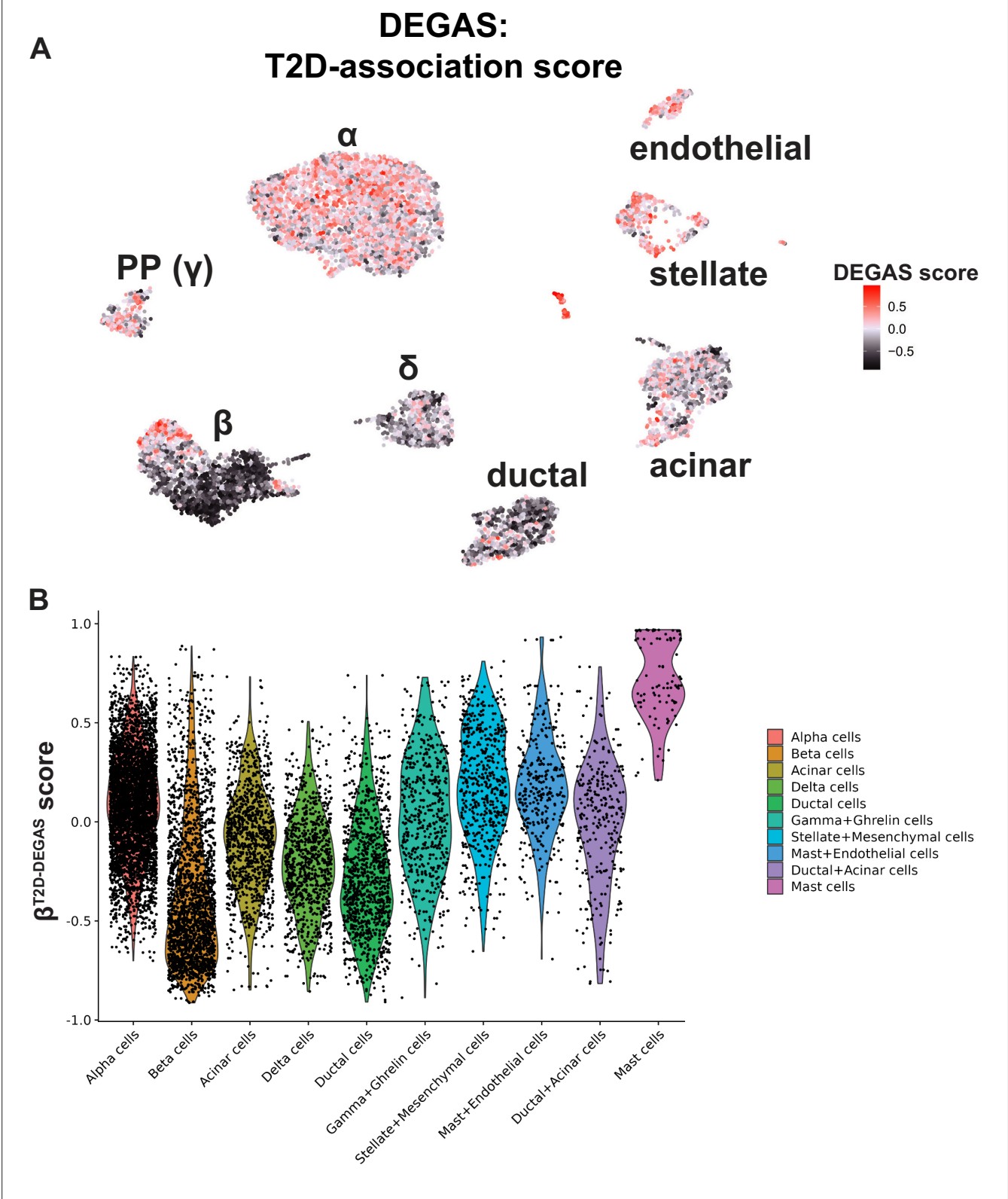

**Figure 3.** DEGAS analysis based on type 2 diabetes (T2D) status. (**A**) DEGAS T2D-association scores (T2D-DEGAS) for each cell were overlaid onto the single-cell UMAP plot. Higher scores in pink/red indicate a strong positive association, and negative scores in darker black indicate a negative association of those cells with T2D. (**B**) Violin plot displaying the aggregate T2D-DEGAS scores per cell type.

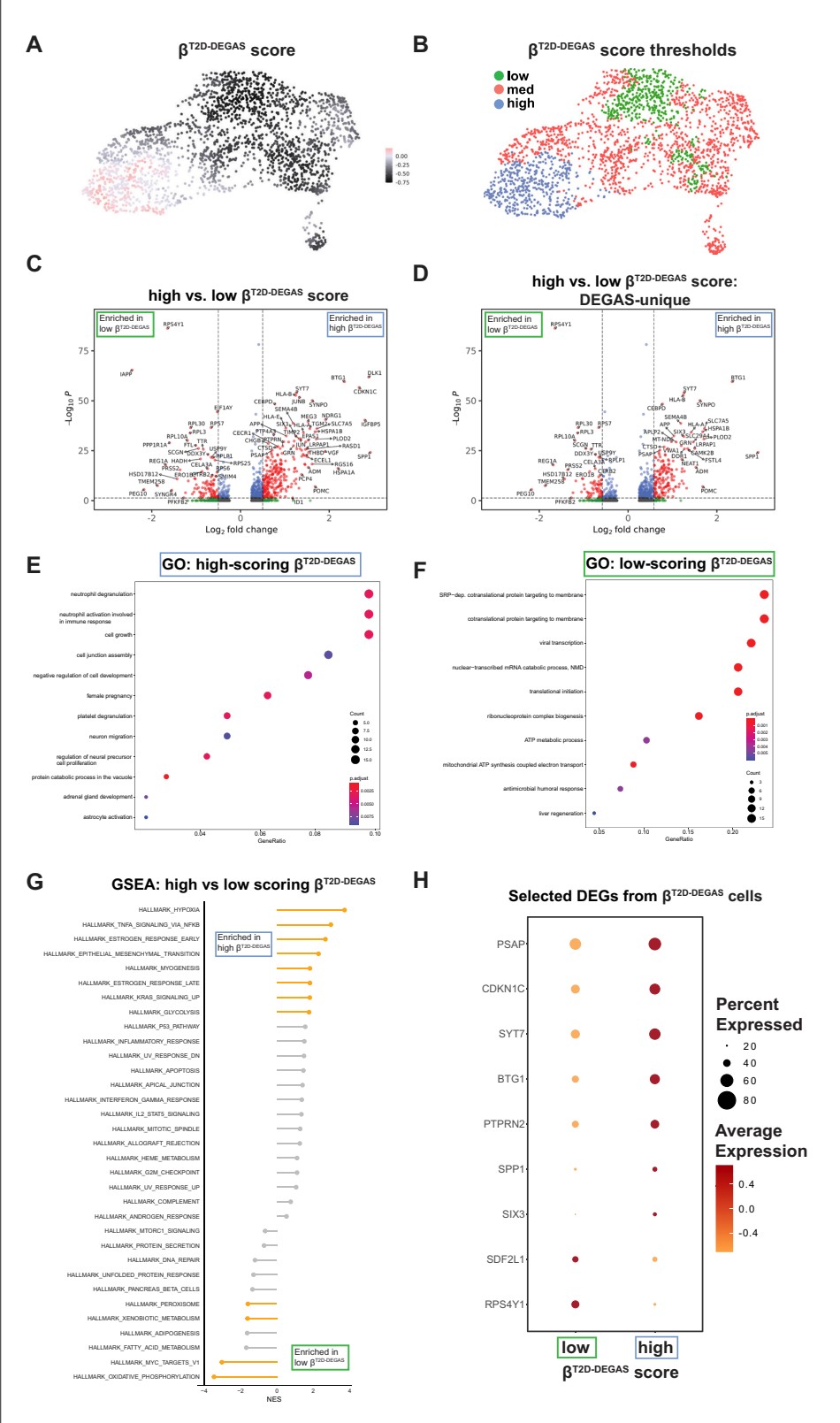

**Figure 4.** Identification of differentially expressed genes in high- and low-scoring $\beta^{T2D-DEGAS}$ cell populations. (**A**) DEGAS scores for type 2 diabetes (T2D) association ($\beta^{T2D-DEGAS}$) were overlaid onto the cell plot. (**B**) β-Cells were classified as low-, medium-, and high-scoring $\beta^{T2D-DEGAS}$ subpopulations for downstream analysis. (**C**) Differential expression analysis of high versus low $\beta^{T2D-DEGAS}$ scores. Genes with p < 0.05 and >1.5 fold change are highlighted

*Figure 4 continued on next page*

*Figure 4 continued*

in red. (**D**) Genes from (**C**) were filtered to remove DEGs that could be identified by comparing β-cells of T2D versus non-diabetic donors in the single-cell data (**Supplementary file 2B**). Gene ontology (GO) analysis of genes enriched in the high (**E**)- and low (**F**)-scoring β$^{T2D-DEGAS}$ subpopulations. (**G**) Gene set enrichment analysis (GSEA) results for high- versus low-scoring β$^{T2D-DEGAS}$ subpopulations. (**H**) Bubble plot highlighting genes driving GO and GSEA categories.

The online version of this article includes the following figure supplement(s) for figure 4:

**Figure supplement 1.** Thresholding analysis for β$^{T2D-DEGAS}$ scores and differential gene expression analysis and plots.

**Figure supplement 2.** Analysis of β$^{obese-DEGAS}$ cells and their functional enrichment.

**2F**) to create ND-β$^{obese-DEGAS}$ and T2D-β$^{obese-DEGAS}$ subpopulations (**Figure 6C**). We next determined differentially expressed genes between the ND-β$^{obese-DEGAS}$ and T2D-β$^{obese-DEGAS}$ cells (**Figure 6D**). GO analysis of genes upregulated in the ND-β$^{obese-DEGAS}$ cluster indicated an enrichment for unfolded protein response (UPR) processes like vesicle transport, translation, and protein folding and stability (**Figure 6E**). Specifically, the ND-βobese-DEGAS cluster was enriched for expression of adaptive (e.g. *MANF* and *HSPA5*) and maladaptive (e.g. *TRIB3* and *DDIT3*) UPR genes, as well as genes involved in ER-associated degradation (e.g. *SEL1L*, *DERL2*, and *SDF2L1*). GO analysis of genes upregulated in the T2D-β$^{obese-DEGAS}$ cluster showed enrichment for hormone transport and secretion pathways (e.g. *IGFBP5*, *UCN3*, *G6PC2*) and inflammatory/immune-related pathways (e.g. *HLA-A/B/E*, *STAT3*) (**Figure 6F**). In parallel, we performed ranked-list GSEA on the ND- and T2D-βobese-DEGAS subpopulations, which was largely consistent with GO results (**Figure 6G**). Significantly enriched pathways for ND-βobese-DEGAS included the UPR and proliferation (E2F and MTORC signaling hallmarks) (**Figure 4—figure supplement 2G**), while T2D-βobese-DEGAS enriched pathways were similar to that of high-scoring β$^{T2D-DEGAS}$ cells and included hypoxia and glycolysis (**Figure 4—figure supplement 2H**). Examples of genes driving the GO and GSEA categories are shown, as well as *CDKN1C* and *DLK1*, given their differential enrichment in both β$^{T2D-DEGAS}$ and T2D-β$^{obese-DEGAS}$ cells (**Figure 6H**).

Combined, our results applying DEGAS using T2D and obesity as disease features support the idea that the identified genes and pathways may underlie heterogeneity in β-cell resilience or susceptibility to obesity-related stress in the context of T2D development. In particular, *DLK1* and *CDKN1C* ranked highly in both T2D-DEGAS and obese-DEGAS analyses (**Figure 4—figure supplement 2I**), making them reasonable candidates for validation of this approach.

## Heterogeneous expression of DLK1 and CDKN1C in human pancreatic islets from diabetic and ND donors with lean, overweight, or obese BMI

To support the DEGAS approach for identifying true markers of β-cell subpopulations or heterogeneity associated with diabetes, we selected DLK1 and CDKN1C as candidate genes for immunostaining in formalin-fixed paraffin-embedded (FFPE) human pancreas sections from ND and T2D donors. We chose Delta-Like 1 Homolog (DLK1) because of its strong associations with T1D (**Wallace et al., 2010**) and T2D (**Kameswaran et al., 2018**). *DLK1* expression overlapped with high-scoring β$^{T2D-DEGAS}$ cells (**Figure 7A**) and with T2D-β$^{obese-DEGAS}$ cells (**Figure 7B**). DLK1 immunostaining primarily colocalized with β-cells in ND human pancreas and exhibited heterogeneity in its expression intensity (**Figure 7C**, **Figure 7—figure supplement 1**). Some islets had DLK1/INS co-staining in many β-cells, while others had only a few DLK1$^+$ β-cells, and this pattern did not appear to be associated with BMI. In T2D pancreas, DLK1 expression appeared to be reduced in some donors; however, an analysis of the distribution of DLK1 staining intensity across INS$^+$ cells indicated that there was no major difference between ND and T2D (**Figure 7D**). We chose our second candidate, Cyclin-Dependent Kinase inhibitor 1C (CDKN1C), because it has been linked to β-cell replication (**Avrahami et al., 2014**). CDKN1C expression also overlapped with high-scoring β$^{T2D-DEGAS}$ cells (**Figure 7E**) and with T2D-β$^{obese-DEGAS}$ cells (**Figure 7F**). CDKN1C staining was primarily nuclear and was heterogeneous across β-cells in ND human pancreas (**Figure 7G**). In agreement with the DEGAS results, CDKN1C staining was increased in β-cell nuclei from T2D donors as quantified by CellProfiler analysis (**Figure 7H**). BMI status did not appear to correlate with CDKN1C expression in either ND or T2D donors.

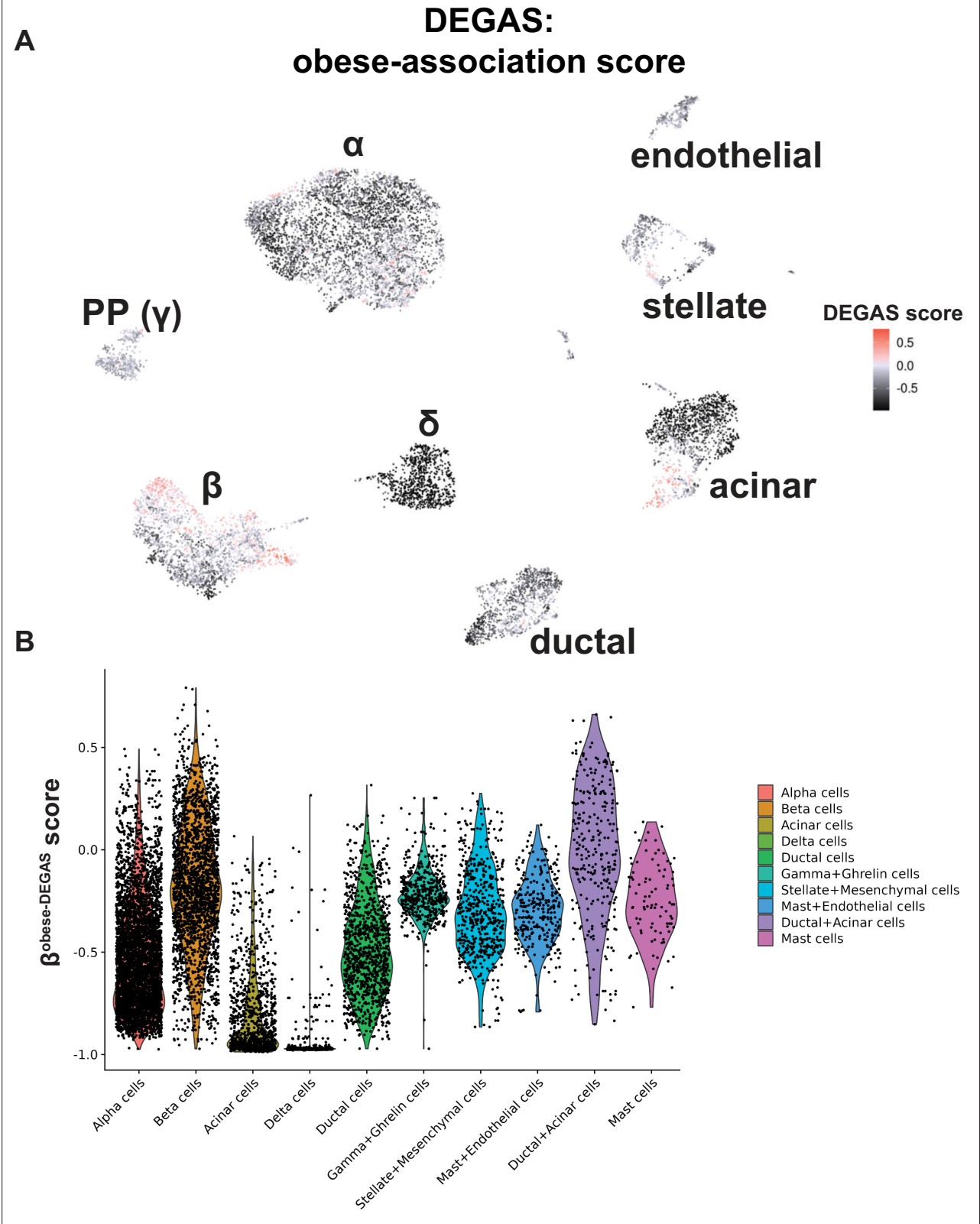

**Figure 5.** DEGAS analysis based on obesity status. (**A**) DEGAS obesity-association scores (obese-DEGAS) for each cell was overlaid onto the single-cell UMAP plot. Higher scores in pink/red indicate a strong positive association, and negative scores in darker black indicate a negative association of those cells with obesity. (**B**) Violin plot displaying the aggregate obese-DEGAS scores per cell type.

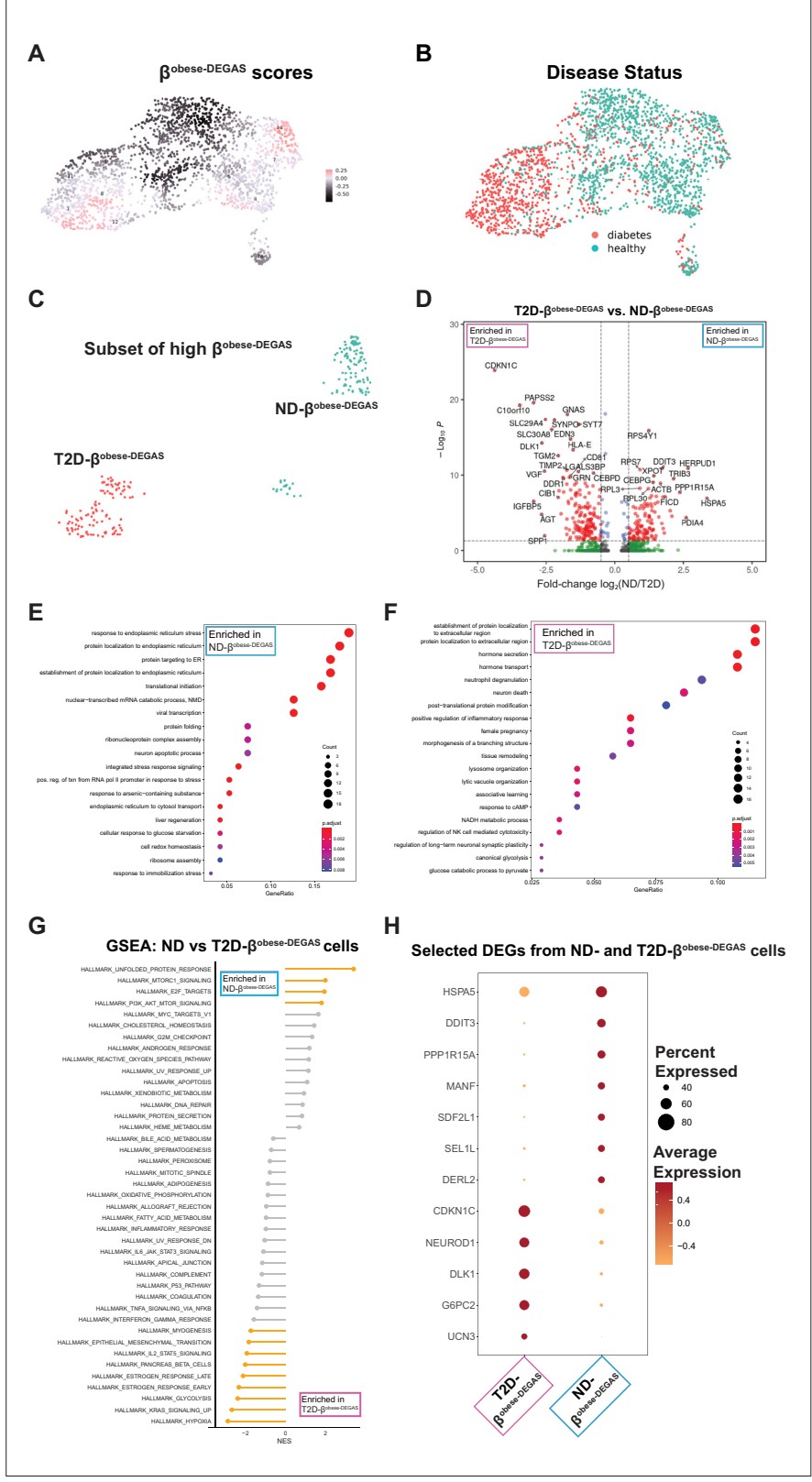

**Figure 6.** Two subpopulations of high β^obese-DEGAS cells defined by underlying single-cell donor type 2 diabetes (T2D) status uncovers stress signature in non-diabetic high-scoring obese-DEGAS β-cells. (**A**) β-Cells were subsetted from *Figure 5A* based on *INS* expression and obese-DEGAS scores overlaid onto the single cells. (**B**) Donor diabetes status overlaid onto the β-cells. (**C**) β-Cells were further subsetted based on both their high obese-

*Figure 6 continued*

DEGAS scores and by the underlying T2D status of the single-cell donors, leading to two major subpopulations, ND-β$^{obese-DEGAS}$ and T2D-β$^{obese-DEGAS}$ cells. (**D**) Differential expression analysis comparing ND-β$^{obese-DEGAS}$ and T2D-β$^{obese-DEGAS}$ subpopulations. Genes with p < 0.05 and >1.5 fold change are highlighted in red. Gene ontology (GO) analysis of genes enriched in the ND-β$^{obese-DEGAS}$ (**E**) and T2D-β$^{obese-DEGAS}$ (**F**) subpopulations. (**G**) Gene set enrichment analysis (GSEA) of ND- versus T2D-β$^{obese-DEGAS}$ subpopulations. (**H**) Bubble plot highlighting genes driving GO and GSEA categories.

## Discussion

### Deep transfer learning is a useful approach to identify disease-associated genes in subsets of heterogeneous islet cells

The rapid advancement of deep learning has unveiled new opportunities in diabetes research, specifically the ability to merge and analyze the large amount of publicly available omics data. The DEGAS deep transfer learning framework provides an extremely powerful and versatile tool allowing for time to event outcomes and classification outcomes to be transferred to individual cells. Notably, this direct transfer to single cells avoids the cluster resolution problems that arise using deconvolution approaches, allowing for subsets of cells within a cluster to be assigned different associations with clinical outcomes. In this way, DEGAS allows the definition of essentially new cell subpopulations that would not necessarily be identified by standard clustering algorithms, and these subpopulations are rationally selected based on assigned disease-association scores. A key point from our study is that the disease-associated single cells identified by DEGAS are those cells whose transcriptomic signature shares complex patterns with patient transcriptomics that have known clinical and metabolic attributes. Because of this, the genes identified as up- or downregulated in subsets of β-cells may be: (1) altered independent of disease due to genetic background of donors; (2) altered downstream of the obese or diabetic states; or (3) representative of resilient or dysfunctional β-cells in the face of metabolic syndrome. Through careful cohort inclusion criteria and batch effect removal, our analyses aim to limit these caveats when possible.

Various deconvolution methods enable estimating relative proportions of cell types in bulk samples using scRNA-seq data (e.g. BSEQ-sc [*Baron et al., 2016*], MuSiC [*Newman et al., 2019*], CIBERSORT [*Newman et al., 2019*], and DWLS [*Tsoucas et al., 2019*]). These methods have been instrumental in identifying changes to relative quantity of cell types within normal and disease tissues; however, they do not provide detailed information about subsets of cell types which may be related to disease and are limited to the resolution of the predefined cell types through clustering. Early attempts to address this include unsupervised learning algorithms (e.g. RaceID) (*Herman and Grün, 2018*). Now, there are multiple advanced computational approaches to identify subsets of cells related to disease status, survival, drug response, and other disease metrics. These tools can be broadly categorized as 'cell prioritization algorithms' and include: DEGAS (*Johnson et al., 2022*), scAB (*Zhang et al., 2022*), Scissor (*Sun et al., 2022*), and scDEAL (*Chen et al., 2022*). Scissor and scAB can assign survival or clinical information from bulk expression data to disparate scRNA-seq datasets using regression and matrix factorization, respectively. scDEAL is a deep learning framework specifically designed for the assignment of drug response information from bulk expression data to subsets of single cells. The major advantages of DEGAS are (1) the ability to select subsets of β-cells at an individual cell resolution based on their disease association; (2) unique qualities of the highly non-linear DEGAS technology versus linear models of other tools to detect complex molecular programs; and (3) speed and scalability based on efficient implementation. DEGAS has the potential to be used on datasets containing well over 1 million cells, unlike other methods.

Nevertheless, certain linear model-based approaches, like RePACT (regressing principal components for the assembly of continuous trajectory), have successfully uncovered genes associated with β-cell heterogeneity in T2D and obesity (*Fang et al., 2019*). In that study, T2D and obesity 'trajectory' genes were identified. We compared the genes enriched in our high-scoring β$^{T2D-DEGAS}$ and in our T2D-β$^{obese-DEGAS}$ cells with the corresponding T2D and obesity trajectory genes from Fang et al., respectively (*Figure 7—figure supplement 2*, *Supplementary file 5*). We noted substantial agreement between DEGAS and RePACT, for example, the transcription factor *SIX3* was enriched in T2D-β$^{obese-DEGAS}$ cells and in Fang et al.'s obese trajectory β-cells (*Fang et al., 2019*). *SIX3* has been implicated as

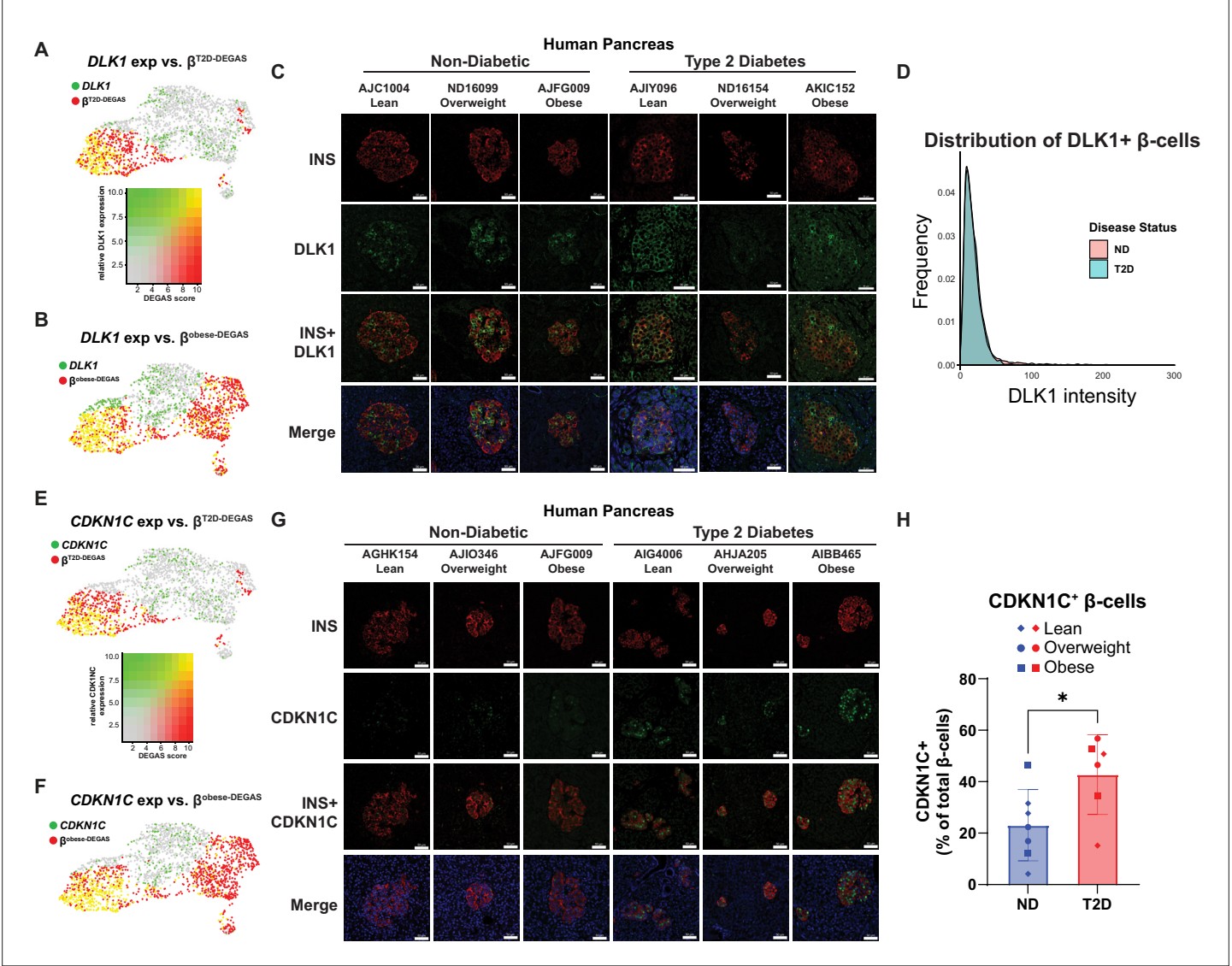

**Figure 7.** DLK1 is a β-cell gene heterogeneously expressed between cells and islets of non-diabetic (ND) and type 2 diabetes (T2D) humans. (**A**) DLK1 expression and β^T2D-DEGAS scores overlaid onto the β-cell cluster. Yellow dots indicate cells with high DLK1 expression and high β^T2D-DEGAS score, as shown in the adjacent overlay heatmap. (**B**) DLK1 expression and β^obese-DEGAS scores overlaid onto the β-cell cluster. Yellow dots indicate cells with high DLK1 expression and high β^obese-DEGAS score. (**C**) Formalin-fixed paraffin-embedded (FFPE) sections of human pancreas from ND and T2D donors were stained for INS and DLK1. Representative images are shown from three different ND and T2D donors. (**D**) CellProfiler analysis showing distribution of DLK1 in ND versus T2D donors. UMAP plot overlaying CDKN1C expression and β^T2D-DEGAS (**E**) or β^obese-DEGAS (**F**) scores. (**G**) FFPE sections of human pancreas from ND and T2D donors were stained for INS and CDKN1C. Representative images are shown from three different ND and T2D donors. (**H**) Increased CDKN1C expression observed in β-cells from T2D patients. *p < 0.05 by Student's *t*-test.

The online version of this article includes the following figure supplement(s) for figure 7:

**Figure supplement 1.** Formalin-fixed paraffin-embedded (FFPE) sections of human pancreas from non-diabetic (ND) and type 2 diabetes (T2D) donors were stained for INS and DLK1.

**Figure supplement 2.** Overlap of RePACT trajectory genes with genes identified by DEGAS in human islet scRNA-seq data.

a diabetes risk gene and is important for β-cell function (*Broadaway et al., 2023*; *Bevacqua et al., 2021*). Additionally, both approaches highlighted the association of *DLK1* with obesity, but only DEGAS identified *DLK1* association with T2D. Overall, the directionality of T2D or obesity gene associations agreed between DEGAS and RePACT results. However, not all the same genes were identified by both approaches; for example, *FXYD2* was identified via RePACT, but not in DEGAS, although *FXYD2* is a downregulated gene in T2D in Marselli et al. and Asplund et al. (*Figure 1G*). Interestingly, *FXYD2*

was was recently shown to be a downstream effector gene of *HNF1A* in single human β-cells (*Weng et al., 2023*). Downregulation of *HNF1A* in T2D β-cells, and reduced *FXYD2* expression as a result, may contribute to membrane hyperpolarization and reduced function (*Weng et al., 2023*). Distinct from the application of RePACT in *Fang et al., 2019*, we did not observe any differences in *HNF1A* expression in our DEGAS analyses, and *FXYD2* was slightly enriched, but not significantly altered, in T2D-β$^{obese-DEGAS}$ cells. Taken together, it is necessary to apply multiple approaches to merged sets of publicly available data because each approach will likely uncover unique and important subsets of β-cells or specific genes.

## Differences between high- and low-scoring β$^{T2D-DEGAS}$ subpopulations

High-scoring β$^{T2D-DEGAS}$ cells had enrichment of pathways like hypoxia and TNFα signaling which contain some overlapping genes like *BTG1*, *BTG2*, and *BHLHE40*. *BHLHE40* was recently shown to cause hypoxia-induced β-cell dysfunction via interfering with *MAFA* expression (*Tsuyama et al., 2023*). *BTG1* and *BTG2* are known as anti-proliferation factors. Given the expression of both *BTG1* and *CDKN1C* in high-scoring β$^{T2D-DEGAS}$ cells, it is possible these cells have altered proliferative potential, even in the context of β-cells which are well known to have low proliferation. Although some genes like *CDKN1C* can be identified as differentially expressed in simple comparisons of all T2D versus ND donor β-cells in scRNA-seq data, DEGAS has the advantage of prioritization or ranking of candidates. For example, *CDKN1C* is within the top ten enriched candidates (by adjusted p-value and fold change) in the β$^{T2D-DEGAS}$ cluster but is 286th when simply comparing all T2D versus ND single β-cells (*Supplementary file 3*). Additionally, deletion of *CDKN1C* has been shown to improve human islet function (*Avrahami et al., 2014*) and gain-of-function mutants are associated with diabetes development and hyperinsulinism (*Kerns et al., 2014*; *Brioude et al., 2015*). In agreement with our results, we found CDKN1C was significantly increased in β-cells from T2D donors (*Figure 7H*). From our analysis, we cannot determine whether this is a response to the diabetic milieu in those donors or if CDKN1C is increased early and prevents β-cell compensation, further contributing to T2D pathogenesis. Understanding the differences between CDKN1C$^+$ and CDKN1C$^-$ human β-cells would be of interest in future work to provide insights into the regulation of β-cell replication.

The low-scoring β$^{T2D-DEGAS}$ cells were enriched in pathways involving oxidative phosphorylation and Myc targets. We also noted enrichment of many small and large ribosomal genes (*RPS/RPL*) in this subpopulation. *RPS/RPL* ribosomal genes are known to be highly expressed in primary human β-cells (*Augsornworawat et al., 2020*), and glucose stimulation induced translation of over 50 different *RPS/RPL* genes in human β-cell line EndoC-βH2 (*Bulfoni et al., 2022*). This suggests that low-scoring β$^{T2D-DEGAS}$ cells may have relatively increased translation and metabolic activity. Taken together, high-scoring β$^{T2D-DEGAS}$ cells may be reflective of dysfunctional β-cells in either a pre-T2D or T2D-onset state, while low-scoring β$^{T2D-DEGAS}$ cells could potentially represent more resilient β-cells.

## Differences between ND-β$^{obese-DEGAS}$ and T2D-β$^{obese-DEGAS}$ subsets of β-cells

Our understanding of why some individuals with obesity develop T2D while others do not is incomplete. A 2021 NHANES report indicated a 41.9% prevalence for obesity in US adults aged 20 and over, with a 14.8% prevalence for diabetes (*Stierman et al., 2021*), in line with CDC reports (*Centers for Disease Control and Prevention, 2022*). This suggests that many individuals with obesity have either not progressed or may never progress to T2D. This phenomenon may be related to those individuals categorized as metabolically healthy obese as compared to metabolically unhealthy obese (*Smith et al., 2019*). Expanded work with DEGAS may uncover specific genes that underlie this relationship. In our analysis, we were able to stratify based on T2D status of the single-cell donors and overlay the relative obesity-association scores from DEGAS. That analysis allowed us to set rational thresholds for grouping subsets of β-cells to compare ND- and T2D-β$^{obese-DEGAS}$ cells. Perhaps counterintuitively, UPR genes were highly enriched in the ND-β$^{obese-DEGAS}$ cells as opposed to the T2D-β$^{obese-DEGAS}$ cells. It is possible these ND-β$^{obese-DEGAS}$ cells are in a stressed pre-diabetic state on the path to T2D. Dominguez-Gutierrez et al. analyzed islet scRNA-seq data using pseudotime analysis and identified three major β-cell states which were defined by their insulin expression and UPR level (*Dominguez-Gutierrez et al., 2019*). The authors speculate that β-cells periodically pass through these states, which in scRNA-seq are visualized as subpopulations of cells. Previously, a *FTH1* subpopulation of β-cells was identified with implications in the UPR (*Muraro et al., 2016*). In agreement with those results, we

found *FTH1* was upregulated in ND-β[obese-DEGAS] cells. Therefore, another possibility is that the UPR induced in ND-β[obese-DEGAS] cells marks a population with lower functionality but higher proliferative potential to combat the insulin-resistant obese state, and the relatively lower UPR in T2D-β[obese-DEGAS] cells possibly correlates with enhanced functionality in established T2D. Concordantly, T2D-β[obese-DEGAS] cells had enrichment of hormone secretion genes (e.g. *SYT7, G6PC2, NEUROD1, UCN3, FFAR1*) in our pathway analysis (*Figure 6F*). In a study of human, mouse, and pig islet scRNA-seq data, subpopulations of 'stressed' β-cells were identified that exhibited enriched hallmark pathways similar to what we observe in ND-β[obese-DEGAS] cells (*Tritschler et al., 2022*). Thus, multiple studies including our current analysis support the existence of subpopulations of stressed β-cells, whether transient or stable, that could in principle be targeted by therapeutics.

We also looked for enrichment of potential secreted biomarkers in high-scoring β[T2D-DEGAS] and T2D-β[obese-DEGAS] cells and identified *LRPAP1* and *C1QL1. LRPAP1* encodes the LDL receptor-related protein-associated protein 1 (LRPAP) and is enriched in this subpopulation. LRPAP is ubiquitously expressed and is predicted to be an ER resident protein (*Bu et al., 1995*). LRPAP appears to be heterogeneously expressed among islet cells and was detected in human plasma in the in the Human Protein Atlas (*Uhlén et al., 2019*). *C1QL1* encodes a secreted peptide that is highly expressed in human islets (*Atanes et al., 2018*) and was enriched in β[T2D-DEGAS] cells, although little else is known about its role in β-cells. Increased C1QL1 release could potentially signal through its GPCR BAI3 to suppress insulin secretion of surrounding β-cells, as has been shown for C1QL3 (*Gupta et al., 2018*).

## Potential for DEGAS in identifying β-cell heterogeneity markers

Subsets or subpopulations of β-cells are an emergent property of β-cell heterogeneity. Two of the most widely used scRNA-seq datasets have also identified significant heterogeneity within pancreatic cell types, notably β-cells (*Segerstolpe et al., 2016*; *Muraro et al., 2016*). Various markers have been identified to define these β-cell subpopulations, including expression of ST8SIA1 and CD9 (*Dorrell et al., 2016*) or Flattop expression (*Bader et al., 2016*). Additionally, UCN3 marks mature β-cells (*van der Meulen et al., 2012*) and RBP4+ β-cells correlated with reduced function (*Camunas-Soler et al., 2020*). Functional heterogeneity has also been described as in the case of leader/first-responder β-cells (*Kravets et al., 2021*; *Salem et al., 2019*) or hub β-cells (*Johnston et al., 2016*), and has been linked to transcription factors including PDX1 (*Weidemann et al., 2024*) and HNF1A (*Weng et al., 2023*). Nevertheless, cellular heterogeneity within a single-cell type is complicated, and there is uncertainty in the single-cell analytics field about what constitutes a stable cell type versus a transient cell state (*Morris, 2019*). Tying transcriptionally distinct clusters to function or morphology may be key to making this distinction (*Clevers et al., 2017*). By inferring disease outcomes in high-resolution cellular subtypes, application of DEGAS can help to inform this debate.

In our analysis, the expression of *DLK1* and *CDKN1C* was enriched in high-scoring β[T2D-DEGAS] and in T2D-β[obese-DEGAS] cells. Recent in vitro work has also highlighted DLK1 and CDKN1C as having increased expression in human pancreatic islets treated with 30 mM glucose for 48 hr (*Yang et al., 2023*). In agreement with scRNA-seq data, we also observed heterogeneous DLK1 and CDKN1C immunostaining of β-cells from human pancreas sections. *DLK1* was previously noted to be enriched in subpopulations of β-cells (*Baron et al., 2016*; *Li et al., 2016*; *Lawlor et al., 2017*; *Mawla and Huising, 2019*; *Xin et al., 2016*); however, its heterogeneity in transcript and protein abundance in ND versus T2D human islet β-cells had not been described until now. DLK1 is a maternally imprinted gene (*Kobayashi et al., 2000*) with described roles in Notch signaling (*Huang et al., 2023*; *Grassi and Pietras, 2022*; *Nueda et al., 2014*) and glucose homeostasis (*Abdallah et al., 2015*). Supporting an active role in β-cell disease, a recent preprint reported DLK1 was required for proper maturation, function, and stress resilience of β-like cells differentiated from human embryonic stem cells (*Zhao et al., 2023*). Our results, combined with the published and available preprint data, suggest the role of DLK1 in β-cell development and function may be nuanced, given that we did not observe a consistent up- or downregulation at the protein level in human islets. However, DLK1 may serve as a surface marker of, or be released from, stressed or at-risk β-cells, but increased sample sizes and in-depth analyses of human islet samples will be required to better describe these possible roles. Our DEGAS results were confirmed; however, CDKN1C expression was increased in T2D β-cells. Other work has shown that suppression of CDKN1C expression in hyperglycemic, immunodeficient mice increased β-cell replication by threefold and those newly replicated cells retained mature β-cell functionality, indicating

that manipulating CDKN1C expression can cause B-cell expansion in T2D patients (*Avrahami et al., 2014*). Additionally, loss-of-function mutations in CDKN1C can cause focal congenital hyperinsulinism, wherein increased β-cell proliferation and IGF-II expression are observed (*Kassem et al., 2001*).

Applying tools like DEGAS to publicly available data will increase our understanding of the underlying β-cell features associated with progression to T2D or with an obese non-T2D state. Expanded DEGAS analyses will be needed to include scRNA-seq data from islets subjected to various models of T1D/T2D (e.g. cytokines, glucolipotoxicity). Major questions include whether the gene candidates identified by DEGAS are protective, disease-causing, or simply markers in β-cell subpopulations. In the future, these questions may be addressed by functional validation studies or observations in genome-wide association studies.

## Limitations of the study and future directions

We think the approach of joining islet omics data to deep learning and artificial intelligence is an area in need of increased attention; however, we appreciate that our current study has limitations. First, our analysis utilized a single bulk RNA-seq human islet dataset, although the Marselli study contains a relatively large number of islet donors compared to most available datasets. Second, we merged only five scRNA-seq human islet datasets for this application of DEGAS. The datasets were chosen because of the extensive meta-analyses to which they have already been subjected (*Mawla and Huising, 2019*), and their use as benchmarking data in many scRNA-seq analysis tools (*Korsunsky et al., 2019*; *Wang et al., 2019*; *Hie et al., 2019*). scRNA-seq has a limitation on the number of detectable genes in each cell, as compared to bulk RNA-seq samples. While these limitations may eventually be mitigated by improved technologies, we can begin to overcome these by merging a larger number of human islet bulk and scRNA-seq studies which contain more donors. Increased sample sizes and study variables come with their own challenges; however, the future gene candidates identified by DEGAS will be of even higher confidence. Additionally, scRNA-seq is a snapshot in time, and it may be tenuous to claim that a particular cell state is transient or persistent. To support a claim that a given subset/subtype of β-cells represents an actual stable population in scRNA-seq data requires at least finding that the cells comprising the candidate population occur across multiple individual donors. In future studies, our use of the diabetes field's considerable investment in islet transcriptomics data and state-of-the-art cell prioritization tools will enable simultaneous identification of disease-associated cellular subtypes and associated biomarkers of function and dysfunction. Additionally, increasing the number of human pancreas donor tissue samples for high-content image analysis will improve confidence in candidate gene validation.

DEGAS has the potential to be applied to even larger mergers of single-cell data (>1 million cells) using the newly developed DEGAS atlas implementations. A vast amount of human (and other species) islet single-cell transcriptomics data is publicly available, but often requires substantial reformatting of metadata and realigning of reads to be harmonized. As our endeavors and those of others proceed, DEGAS will lead to even higher confidence predictions of β-cell subpopulations. It is also apparent from our DEGAS analyses that other non-β-cells are highlighted in the islet single-cell map for associations with T2D and obesity. Although outside the scope of this work, further exploration of our current DEGAS analyses, or of analyses using larger single-cell integrations, will have implications for the other major islet and pancreas cell types, including α-cells, δ-cells, and PP-cells. For example, PP-cells have been shown to have a role in pancreatogenic diabetes as opposed to T2D (*Hart et al., 2023*), and there appear to be subpopulations of high- and low-scoring PP[T2D-DEGAS] cells within the DEGAS analysis (*Figure 3A*). Our successful application of deep transfer learning in human islet data opens the possibility of predicting subtypes of β-cells in other diseases, like T1D and congenital hyperinsulinism, to find potential biomarkers and therapeutic candidates. In some rare diseases, like congenital hyperinsulinism, single-cell data is not available; however, spatial transcriptomics (ST) could be applied to FFPE sections. DEGAS has already been implemented to analyze ST data in prostate cancer and diabetic pancreas (*Alsaleh et al., 2022*; *Couetil et al., 2023*; *Chatterjee et al., 2025*), and will be increasingly applicable to human pancreas samples as ST technology continues to advance. It is important to note that DEGAS is just one of many machine learning approaches to analyze transcriptomic data. It will be important to include DEGAS in combination with both linear and other non-linear models to capture as many relevant gene candidates as possible.

## Methods

### Human islet bulk transcriptomic dataset acquisition, processing, and analysis

In this study, human islet bulk RNA-seq raw count data (aligned to GRCh38) and donor metadata from Marselli et al. GSE159984 was downloaded from the Gene Expression Omnibus (*Marselli et al., 2020*; *Supplementary file 1*). The rationale for using this dataset was its large sample size of both ND and T2D samples and the agreement of differentially expressed genes between the GSE159984 dataset and a similarly large independent dataset (*Asplund et al., 2022*). All genes in the read count data were filtered based on the one-to-one identifier correspondences between gene symbol, Entrez gene identifier, and Ensembl identifier from the Matched Annotation from NCBI and EBI table (MANE GRCh38 v1.1) (*Morales et al., 2022*). Sample data were labeled and grouped by their RRID and disease status (ND vs. T2D). After filtering, the GSE159984 read count table contained 19,058 genes from $N$ = 27 T2D samples and $N$ = 58 non-diabetes samples (*Figure 1A*). Differential gene expression analysis between ND and T2D groups was performed using the edgeR likelihood ratio test and cutoffs were FDR <0.05 and $|\log_2 \text{fold-change}| \geq 0.58$. The processed and filtered read count table and edgeR results are provided in *Supplementary file 2* and the R script used for processing is available on GitHub (https://github.com/kalwatlab/Islet_DEGAS_v1, copy archived at *Kalwat, 2025*). Selected individual genes are shown in *Figure 1H* as box plots to illustrate the quartiles across all donor samples.

### Human islet single-cell data acquisition, filtering, and integration

We obtained read count tables for five single-cell human islet datasets (GSE84133; GSE85241; E-MTAB-5061; GSE81608; GSE86469 *Supplementary file 1*) which were realigned to GRCh38.p5 (hg38) (*Mawla and Huising, 2019*; *Figure 1B*). Metadata were downloaded from GEO or obtained from the supplemental information of the respective publications. These five datasets were previously integrated and analyzed (*Mawla and Huising, 2019*). To prepare the data for our machine learning analyses, we repeated the integration and analysis of these datasets in R using Seurat to exclude cells with low expressed genes and cells with over 20% mitochondrial gene expression (low viability). The upper and lower limits varied between the datasets to account for differences in library preparation and sequencing platform following guidelines in the Seurat documentation.

Datasets were integrated into a single dataset using Seurat version 4.1.3, the Seurat objects were normalized using regularized negative binomial regression (SCTransform) to correct for batch effect and clustered using default settings. The most variable features were used to identify integration anchors which are passed to the *IntegrateData* function to return a Seurat object containing an integrated expression matrix of all cells.

The clusters were visualized using UMAP, and expression of pancreatic hormone genes INS, GCG, SST, PPY, and GHRL was used to annotate β, α, delta, γ, and ε cell clusters, respectively. Clustering analysis used the Louvain algorithm (*Blondel et al., 2008*) and labeling was performed to identify different cell types because our integrated dataset contained a mixture of cells from pancreatic islets. In total, 22 clusters were identified from 17,273 pancreatic cells, and the clusters were classified into endocrine and non-endocrine cell types based on cell-specific marker expression (*Figure 2A*). R scripts are available on GitHub (https://github.com/kalwatlab/Islet_DEGAS_v1, *Kalwat, 2025*). The Seurat object containing all read count matrices and merged data and metadata is available on Mendeley (*Kalwat, 2024*).

### Transfer learning using DEGAS on human islet transcriptomic data

The integrated single-cell and bulk datasets were processed with the DEGAS (v1.0) pipeline (https://github.com/tsteelejohnson91/DEGAS, *Johnson and Huang, 2024*; *Johnson et al., 2022*) to calculate disease risk scores associated with T2D status or BMI status. The bulk expression data were scaled and normalized prior to DEGAS analysis using the *preprocessCounts* function. The merged scRNA-seq expression matrix, the bulk expression matrix, and donor sample labels (matched with the bulk samples) were used as input. The intersection of highly variable genes between scRNA-seq and bulk expression data was used for further analysis. The DEGAS model was trained and predicted on the formatted data. Updated scripts and instructions for running DEGAS are available on GitHub

(version 1.0) (https://github.com/tsteelejohnson91/DEGAS, *Johnson and Huang, 2024*). DEGAS for T2D and BMI were run independently.

In DEGAS for T2D, the donor labels were 'normal' versus '2D'. In DEGAS for BMI metadata, donors from bulk RNA-seq were categorized and labeled as 'lean' (<25 BMI), 'overweight' (25–30 BMI), and 'obese' (>30 BMI) according to CDC guidelines. The models were trained to calculate T2D- and BMI-association scores for each of the respective categories. We identified differentially expressed genes using the *FindMarkers* function in β-cells with high versus low T2D-association scores ($\beta^{T2D-DEGAS}$) or high obesity scores ($\beta^{obese-DEGAS}$) among healthy or T2D donor single cells.

## Subclustering of β-cells and post-DEGAS scRNA-seq analysis

We isolated the β-cell cluster using *subset* function and reclustered them separately from the other islet cell types. The β-cells were further classified by different thresholding parameters including median and quantiles. For quantile and median thresholds, we generated high, medium, and low, as well as upper 50% and lower 50% of median. We used the *FindMarkers* function to identify differentially expressed genes in high versus low (*Figure 4D*), high versus medium + low, and median (upper 50% vs. lower 50%) (*Supplementary file 3*).

## Pathway analysis

To analyze differentially expressed genes that were either up- or downregulated between clusters, enrichment analyses and plot generation were completed using ClusterProfiler, EnhancedVolcano, and ggplot2 R packages in R version 4.2.2 and RStudio. Enrichment of gene sets within the Biological Process GO terms is shown, and cluster profiler outputs are provided in *Supplementary file 4*. Venn diagrams for comparing DEGAS and RePACT hit genes (*Supplementary file 5*) were created using https://molbiotools.com/listcompare.php. GSEA was run using GSEA software (*Mootha et al., 2003*; *Subramanian et al., 2005*) downloaded from https://www.gsea-msigdb.org/gsea/ and the MSigDB Hallmark Gene Set (*Liberzon et al., 2015*) from the Broad Institute. For bulk RNA-seq analysis, the edgeR results containing all expressed genes were used as input to run standard GSEA. For GSEA analysis of comparisons between subpopulations of β-cells from scRNA-seq data, a ranked list based on $\log_2$fold-change was generated without a cutoff for adjusted p-value. GSEAPreranked mode used the 'classic' enrichment statistic for 1000 permutations.

## Human pancreas tissue staining and microscopy

FFPE deidentified human pancreas tissue was obtained through the Integrated Islet Distribution Program (IIDP) and the National Disease Research Interchange (NDRI) (*Supplementary file 6*). IIDP provided 5 µm tissue sections on slides, and NDRI tissue blocks were processed into 5 µm sections and mounted on glass slides by the Indiana University School of Medicine Histology Lab Service Core. Slides were deparaffinized by xylene and ethanol washes. Antigen retrieval was performed by heating for 40 min in an Epitope Retrieval Steamer with slides submerged in Epitope Retrieval Solution (IHC-Tek). Subsequently, slides were placed onto disposable immunostaining cover plates and inserted into the Sequenza slide rack (Ted Pella/EMS) for washing, blocking, and antibody incubations. After three 10 min washes in IHC wash buffer (0.1% Triton X-100 and 0.01% sodium azide in PBS pH 7.4), slides were blocked for 1 hr at room temperature in normal donkey serum (NDS) block solution (5% donkey serum, 1% bovine serum albumin, 0.3% Triton X-100, 0.05% Tween-20, and 0.05% sodium azide in PBS pH 7.4). Slides were then incubated overnight at 4°C with primary antibodies diluted in NDS block solution (*Supplementary file 7*). After three washes in IHC wash buffer 200 µl each, slides were incubated in secondary antibodies in NDS block solution for 1 hr at room temperature, followed by DAPI staining for 10 min and three additional washes. The washed slides were mounted in polyvinyl alcohol (PVA) mounting medium (5% PVA, 10% glycerol, 50 mM Tris pH 9.0, 0.01% sodium azide) or Prolong Gold and imaged on Zeiss LSM710 confocal microscope equipped with a Plan-Apochromat 20x/0.8 objective (#420650-9901). Images were processed in the Zeiss Zen software to add scale bars, set coloration for channels, and generate merged images. Scale bars indicate 50 µm. Distribution of DLK1 intensity within $INS^+$ areas and $CDKN1C^+$ β-cell nuclei was quantified using CellProfiler (v4.2.7). Data were processed in R and plotted in GraphPad Prism.

## Statistical analysis

In edgeR bulk RNA-seq differential gene expression analysis, the likelihood ratio test was applied. In the FindMarkers function in Seurat, the default Wilcoxon Rank Sum test was used for differential gene expression analysis. Cutoffs for significant differentially expressed genes were set at $|log_2fold\text{-}change| > 0.58$ and adjusted p-value <0.05.

## Declarations

### Consent for publication

Human islet transcriptomic data used in this project are all published datasets and the donors are deidentified. Human pancreatic tissue sections were obtained from NDRI and were deidentified. Studies on deidentified human tissue or publicly available human transcriptomics datasets do not constitute human research as defined in 45 CFR 46.102(f).

## Acknowledgements

We thank Dr. Mark Huising for helpful conversations and providing access to processed data. Thank you to Dr. Andrew Templin for helpful conversations and critical review of the manuscript. Thank you to Dr. Anthony Piron at Université Libre de Bruxelles for assistance with metadata mapping for GSE159984. This work was supported by the Histology Core of the Indiana Center for Musculoskeletal Health at IU School of Medicine and the Bone and Body Composition Core of the Indiana Clinical Translational Sciences Institute (CTSI). A specific thanks to Drew Brown in the Histology Lab Service Core for assisting with pancreas sectioning. This work was supported by internal funds at the Indiana Biosciences Research Institute (MAK and TSJ), AnalytixIN (TSJ), Indiana University Precision Health Initiative (TSJ), 1R01GM148970 (TSJ), 1R21CA264339 (TSJ).

## Additional information

### Funding

| Funder | Grant reference number | Author |
|---|---|---|
| National Institutes of Health | 1R01GM148970 | Travis S Johnson |
| National Institutes of Health | 1R21CA264339 | Travis S Johnson |

The funders had no role in study design, data collection, and interpretation, or the decision to submit the work for publication.

### Author contributions

Gitanjali Roy, Conceptualization, Data curation, Formal analysis, Validation, Investigation, Visualization, Methodology, Writing – original draft, Writing – review and editing; Rameesha Syed, Olivia Lazaro, Sylvia Robertson, Alex M Mawla, Data curation, Formal analysis; Sean D McCabe, Supervision, Writing – review and editing; Daniela Rodriguez, Data curation, Formal analysis, Validation, Visualization; Travis S Johnson, Conceptualization, Software, Formal analysis, Supervision, Funding acquisition, Writing – original draft, Project administration, Writing – review and editing; Michael A Kalwat, Conceptualization, Data curation, Formal analysis, Supervision, Funding acquisition, Validation, Investigation, Visualization, Writing – original draft, Project administration, Writing – review and editing

### Author ORCIDs

Gitanjali Roy ⓘ https://orcid.org/0000-0002-9622-0184
Travis S Johnson ⓘ https://orcid.org/0000-0002-4628-2256
Michael A Kalwat ⓘ https://orcid.org/0000-0002-8349-9470

Reviewer #1 (Public review): https://doi.org/10.7554/eLife.96713.3.sa1
Reviewer #2 (Public review): https://doi.org/10.7554/eLife.96713.3.sa2

Author response https://doi.org/10.7554/eLife.96713.3.sa3

## Additional files

### Supplementary files

Supplementary file 1. List of bulk and single-cell transcriptomic datasets.

Supplementary file 2. Bulk RNA-seq metadata, analysis, and comparisons to publicly available datasets.

Supplementary file 3. Differential gene expression in DEGAS-identified β-cell subpopulations.

Supplementary file 4. Pathway analysis results.

Supplementary file 5. Comparison of DEGAS T2D and obesity genes to Fang et al. RePACT trajectory genes.

Supplementary file 6. Formalin-fixed paraffin-embedded (FFPE) human pancreas metadata.

Supplementary file 7. Antibodies.

MDAR checklist

### Data availability

The raw datasets supporting the results of this article are available in the NCBI GEO and the Array-Express repositories under the following persistent identifiers (also shown in *Supplementary file 1*): bulk RNA-seq: GSE159984; scRNA-seq: GSE84133, GSE85241, GSE86469, E-MTAB-5061, GSE81608. Scripts for bulk and single-cell RNA-seq analysis and DEGAS analysis are available at GitHub (copy archived at *Kalwat, 2025*).

The following previously published datasets were used:

| Author(s) | Year | Dataset title | Dataset URL | Database and Identifier |
|---|---|---|---|---|
| Marchetti P, Piron A, Suleiman M, Colli M, Yi X, Khamis A, Rutter G, Bugliani M, Giusti L, Ronci M, Ibberson M, Turatsinze J, Boggi U, De Simone P, Tata De, De Luca C, Campani D, Lopes M, Nasteska D, Shulte A, Solimena M, Hecht P, Rady B, Bakaj I, Pocai A, Norquay L, Thorens B, Canouil M, Frogues P, Eizirik D, Cnop M, Marselli L | 2020 | Persistent or Transient Human β-cell Dysfunction Induced by Metabolic Stress Associated with Specific Signatures and Shared Gene Expression of Type 2 Diabetes | https://www.ncbi.nlm.nih.gov/geo/query/acc.cgi?acc=GSE159984 | NCBI Gene Expression Omnibus, GSE159984 |
| Veres A, Baron M | 2016 | A single-cell transcriptomic map of the human and mouse pancreas reveals inter- and intra-cell population structure | https://www.ncbi.nlm.nih.gov/geo/query/acc.cgi?acc=GSE84133 | NCBI Gene Expression Omnibus, GSE84133 |
| Muraro MJ, Dharmadhikari G, de Koning E, van Oudenaarden A | 2016 | A single-cell transcriptome atlas of the human pancreas [CEL-seq2] | https://www.ncbi.nlm.nih.gov/geo/query/acc.cgi?acc=GSE85241 | NCBI Gene Expression Omnibus, GSE85241 |
| Xin Y, Gromada J | 2016 | RNA Sequencing of Single Human Islet Cells Reveals Type 2 Diabetes Genes | https://www.ncbi.nlm.nih.gov/geo/query/acc.cgi?acc=GSE81608 | NCBI Gene Expression Omnibus, GSE81608 |

*Continued on next page*

*Continued*

| Author(s) | Year | Dataset title | Dataset URL | Database and Identifier |
|---|---|---|---|---|
| Palasantza A, Sandberg R, Segerstolpe A | 2016 | Single-cell RNA-seq analysis of human pancreas from healthy individuals and type 2 diabetes patients | https://www.ebi.ac.uk/biostudies/arrayexpress/studies/E-MTAB-5061 | ArrayExpress, E-MTAB-5061 |
| Lawlor N, George J, Bolisetty M, Kursawe R, Sun L, Sivakamasundari V, Kycia I, Robson P, Stitzel ML | 2016 | Single cell transcriptomics defines human islet cell signatures and reveals cell-type-specific expression changes in type 2 diabetes [single cell] | https://www.ncbi.nlm.nih.gov/geo/query/acc.cgi?acc=GSE86469 | NCBI Gene Expression Omnibus, GSE86469 |

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
