## [Editor Report · eLife assessment]

This is a **useful** study that applies deep transfer learning to assign patient-level disease attributes to single cells of T2D and non-diabetic patients, including obese patients. This analysis identified a single cluster of T2D-associated β-cells; and two subpopulations of obese- β-cells derived from either non-diabetic or T2D donors. The findings were validated at the protein level using immunohistochemistry on islets derived from non-diabetic and T2D organ donors, contributing **solid** experimental evidence for the computational analyses.

---

## [Referee Report · Reviewer #1 (Public review)]

In this manuscript, Roy et al. used the previously published deep transfer learning tool, DEGAS, to map disease associations onto single-cell RNA-seq data from bulk expression data. The authors performed independent runs of DEGAS using T2D or obesity status and identified distinct β-cell subpopulations. β-cells with high obese-DEGAS scores contained two subpopulations derived largely from either non-diabetic or T2D donors. Finally, immunostaining using human pancreas sections from healthy and T2D donors validated the heterogeneous expression and depletion of DLK1 in T2D islets.

Strengths:

(1) This meta-analysis of previously published scRNA-seq data uses a deep transfer learning tool.

(2) Identification of novel beta cell subclusters.

(3) Identified a relatively innovative role of DLK1 in T2D disease progression.

Comments on revisions:

All previous concerns have been addressed.

---

## [Referee Report · Reviewer #2 (Public review)]

Summary:

The manuscript by Gitanjali Roy et al. applies deep transfer learning (DEGAS) to assign patient-level disease attributes (metadata) to single cells of T2D and non-diabetic patients, including obese patients. This led to the identification of a singular cluster of T2D-associated β-cells; and two subpopulations of obese- β-cells derived from either non-diabetic or T2D donors. The objective was to identify novel and established genes implicated in T2D and obesity. Their final goal is to validate their findings at the protein level using immunohistochemistry of pancreas tissue from non-diabetic and T2D organ donors.

Strengths:

This paper is well-written, and the findings are relevant for β-cell heterogeneity in T2D and obesity.

Weaknesses:

The validation they provide is not sufficiently strong: no DLK1 immunohistochemistry is shown of obese patient-derived sections. Additional presumptive relevant candidates from this transcriptomic analysis should be screened for, at the protein level.

Comments on revisions:

The authors have largely addressed my comments. No further experiments are requested.

---

## [Author Response]

The following is the authors’ response to the original reviews.

**Reviewer #1 (Public Review):**
In this manuscript, Roy et al. used the previously published deep transfer learning tool, DEGAS, to map disease associations onto single-cell RNA-seq data from bulk expression data. The authors performed independent runs of DEGAS using T2D or obesity status and identified distinct β-cell subpopulations. β-cells with high obese-DEGAS scores contained two subpopulations derived largely from either non-diabetic or T2D donors. Finally, immunostaining using human pancreas sections from healthy and T2D donors validated the heterogeneous expression and depletion of DLK1 in T2D islets.Strengths:(1) This meta-analysis of previously published scRNA-seq data using a deep transfer learning tool.(2) Identification of novel beta cell subclusters.(3) Identified a relatively innovative role of DLK1 in T2D disease progression.Thank you for your comments on the strengths of our work.Weaknesses :“There is little overlap of the DE list of bulk RNA-seq analysis in Figure 1D and 1E overlap with the DE list of pseudo-bulk RNA-seq analysis of all cells in Figure S2C. “

Thank you for pointing this out. To clarify, we did not perform pseudo-bulk analysis on the scRNAseq data. Instead, we used the Seurat FindClusterMarkers function to identify differentially enriched genes between T2D and ND single cells. Indeed, there are many significant genes in new Fig S2D (original S2C). There is some overlap between those data and the DEGS from bulk RNAseq data in Fig 1D, including IAPP, ENTPD3, and FFAR4. However, the limited overlap supports the notion that improved approaches are necessary to identify candidate DEGs from single cell data, as simply performing a comparison of T2D to ND of all β-cells may miss important genes or include many false positives. We have now added clarification to the text to highlight this point.

The biological meaning of "beta cells had the lowest scores compared to other cell types" is not clear.

The relatively lower T2D-DEGAS scores for beta cells overall compared to all other cell types (alpha cells, acinar cells, etc) likely reflects the fact that in T2D, beta cell-specific genes can be downregulated. This affects the DEGAS model which is reflected in the scores of all cells in the scRNAseq data. By subsetting the beta cells and replotting them on their own, we can analyze the relative differences in DEGAS scores between different subsets of beta cells. We have now amended the text to clarify, as follows:

“We next mapped the T2D-association scores onto the single cells (Fig 3A). β-cells had a wide distribution of scores, possibly reflecting β-cell heterogeneity or altered β-cell gene expression after onset of T2D (Fig 3B).”

The figures and supplemental figures were not cited following the sequence, which makes the manuscript very difficult to read. Some supplemental figures, such as Figures S1C-S1D, S2B-S2E, S3A-S3B, were not cited or mentioned in the text.

We apologize for this oversight and have now amended the text to call out all figures/panels in order of first introduction.

In Figure 7, the current resolution is too low to determine the localization of DLK1.

We have confirmed that in our Adobe Illustrator file, each microscopy panel has a DPI of >600. We have also provided the highest quality TIFF file versions of our figure set. We hope the reviewer will have access to download the high-quality TIFF file for Fig 7 if possible, or the editorial staff can provide it.

As a result of addressing the critiques, we identified CDKN1C as another promising candidate enriched in the β^T2D-DEGAS^ and β^obese-DEGAS^ subpopulations of β-cells. We found that CDKN1C is heterogeneously expressed at the protein level in β-cells and that it is increased in T2D in agreement with the DEGAS predictions. We have amended the manuscript to highlight CDKN1C more prominently while still discussing DLK1. DLK1 is very interesting, but exhibits greater donor to donor variability in its alterations in T2D.

**Reviewer #2 (Public Review):**
Summary:The manuscript by Gitanjali Roy et al. applies deep transfer learning (DEGAS) to assign patient-level disease attributes (metadata) to single cells of T2D and non-diabetic patients, including obese patients. This led to the identification of a singular cluster of T2D-associated β-cells; and two subpopulations of obese- β-cells derived from either non-diabetic or T2D donors. The objective was to identify novel and established genes implicated in T2D and obesity. Their final goal is to validate their findings at the protein level using immunohistochemistry of pancreas tissue from non-diabetic and T2D organ donors.Strengths:This paper is well-written, and the findings are relevant for β-cell heterogeneity in T2D and obesity.

Thank you for your comments on the positive aspects of our work.

Weaknesses:The validation they provide is not sufficiently strong: no DLK1 immunohistochemistry is shown of obese patient-derived sections.

We have acquired additional FFPE pancreas samples from the Integrated Islet Distribution Program (IIDP) from lean, overweight, and obese humans with and without T2D. We have now stained for CDKN1C and DLK1 in these samples and have integrated the data into Fig 7 and Fig S5.

Because the data with CDKN1C was more striking and consistent with the DEGAS predictions, we have chosen to highlight CDKN1C in the main figure and text. The DLK1 data is still quite interesting, although there is substantial variability between T2D donors when it comes to altered staining intensity. DLK1 presents an interesting challenge, given multiple isoforms and cleavage products, and will require further investigation as the focus of a different manuscript.

Additional presumptive relevant candidates from this transcriptomic analysis should be screened for, at the protein level.

Thank you for this suggestion. We also identified CDKN1C as promising candidate enriched in the β^T2D-DEGAS^ and β^obese-DEGAS^ subpopulations of β-cells. We found that CDKN1C is heterogeneously expressed at the protein level in β-cells and that it is increased in T2D in agreement with the DEGAS predictions. We have amended the manuscript to highlight CDKN1C more prominently while still discussing DLK1. DLK1 is very interesting but exhibits greater donor to donor variability in its alterations in T2D.

**Reviewer #1 (Recommendations For The Authors):**
Please explain and provide the detailed information on what percentage of the DE list of bulk RNA-seq analysis in Figures 1D and 1E overlap with the DE list of pseudo-bulk RNA-seq analysis of all cells in Figure S2C.

Addressed in response to R1 Comment 1.

Please provide the definition of each cluster of UMAP of the merged human islet scRNA-seq data.

In figure panels 2A-B,D-G and 3A, the clusters are now labeled according to the marker genes described in Fig 2C.

The integrative UMAP needs to be included in the main figure.

We have now moved previous Fig S2A and S2B into the main figures as new Fig 2A-B.

All figures and supplemental figures need to be cited following sequence.

Addressed in response to R1 Comment 3.

In Figure 7, high-resolution images are needed to determine the colocalization of INS and DLK1.

Addressed in response to R1 Comment 4.

**Reviewer #2 (Recommendations For The Authors):**
Results: 124-128: Fig 1H_The error bars seem high, please include whether the boxplots are SEM or SD. Also, more detail on statistics is missing.

Thank you for pointing out the need for clarification here. The whiskers on the box and whiskers plots are not error bars. By default, in geom_boxplot() and stat_boxplot(), the whiskers extend to 1.5 times the interquartile range. The box itself represents 50% of the data, the bottom of the box is the first quartile, the middle horizontal line is the median, and the top line of the box is the third quartile. We have now added a clearer description of this to the figure legend and in the methods section.

The genes shown in Fig 1H were selected because they are found in the T2D Knowledge Portal, illustrating a clear link to T2D. At the T2DKP (https://t2d.hugeamp.org/research.html?pageid=mccarthy_t2d_247), PAX4 and APOE are listed as causal, SLC2A2 has strong evidence, and CYTIP has a linked SNP. This is now discussed in the results section before the Fig 1H callout. These genes are significantly differentially expressed using edgeR in panel 1D with FDR<0.05. The individual data points for each human are shown.

Figure 6: In general, the representation of the data is quite misleading. It would be nice to have an alternative way of presenting the data, especially when comparing beta-obese differentially expressed genes and pathways and T2D beta obese. Maybe an additional Venn diagram can help. Also, it would be nice to compare data from T2D beta nonobese to ND beta obese, especially given how the story is presented in the paper.

Thank you for pointing out this clarity issue. We agree that additional alternate ways to present the data would be helpful. When we performed DEGAS using BMI as the disease feature we noted two major and one minor clusters of high-scoring cells in Fig 6A .

**Author response image 2. sa3fig2:** 

This contrasted with the score map when we ran DEGAS with T2D as the disease feature

The main difference seems to be the low scoring β^T2D-DEGAS^ cluster is different from the low β^obese-DEGAS^ cluster.

Therefore, we could not easily apply thresholding to the β^obese-DEGAS^ scores, so instead we subsetted them for comparison. It was also apparent from the metadata that single cells from the left-hand side of the β-cell cluster came from donors that had T2D.

To clarify these points and address the reviewer’s concerns, we have added a comparison of the DEGs identified for β^T2D-DEGAS^ high vs. low and T2D-β^obese-DEGAS^ vs ND-β^obese-DEGAS^ in Fig S4J, also shown below. DLK1 and CDKNC1C fall within the intersection, in addition to being two of the most enriched candidates in each DEGAS run (Fig 4C and Fig 6D).

220-222: Figure 7C_ Is one of the nondiabetic beta samples obese? If so, please clearly label it; if not, that info is missing. One would expect that the DLK1 expression in ND obese beta cells resembles the T2D beta cell and not ND non-obese beta cells. That's a big point of this entire work, and experimentally missing. Additional candidate proteins should be checked.

We have amended the entire Fig 7 to include more data for DLK1 staining as well as adding staining for CDKN1C. We also used CellProfiler to quantify the intensity distribution of DLK1 staining in β-cells and overall found that our initial conclusions were not supported when considering an increased sample size. DLK1 expression is heterogeneous both within and between donors. While we have data from T2D donors that shows DLK1 is lost, other T2D samples indicate that DLK1 is not always lost. At least in the current sample set we have analyzed, we cannot conclude that there is a clear correlation between diabetes or BMI for DLK1. Why DLK1 labels some β-cells and not others and what the role of this subpopulation is an open question.

Alternatively, we greatly appreciate the reviewer’s suggestion to validate additional candidates, as this led us to CDKN1C. In new Fig 7E-H we now show that CDKN1C is increased in T2D β-cells, in agreement with the DEGAS predictions.

This work shows that machine learning approaches are powerful for identifying potential candidates, but it also highlights the need for these predictions to be validated at the protein level in human samples.

Discussion: Based on lack of supporting IHC data, this is an overstatement:“DLK1 expression highly overlapped with high scoring βT2D DEGAS cells (Figure 7A) and with T2D βobese-DEGAS cells (Figure 7B). DLK1 immunostaining primarily colocalized with β-cells in non-diabetic human pancreas (Figure 7C). DLK1 showed heterogeneous expression within islets and between islets within the same pancreas section, wherein some islets had DLK1/INS co-staining in most β-cells and other islets had only a few DLK1+ β-cells. In the T2D pancreas, DLK1 staining was much less intense and in fewer β-cells, yet DLK1+/INS+ cells were observed (Figure 7C). This contrasts with the relatively higher DLK1 gene expression seen in the β-cells from the βT2D-DEGAS and T2D-βobese-DEGAS subpopulations (Figure 4D & 6C) as highlighted in Figure 7A,B. which were up- or down-regulated in subpopulations of β-cells identified by DEGAS, and to validate our findings at the protein level using immunohistochemistry of pancreas tissue from non-diabetic and T2D organ donors.”

This part was at the very end of the last results subsection. This section has been largely rewritten to better describe the new figure and the language has been tempered to not overinterpret the data shown.

“Our current findings applying DEGAS to islet data have implications for β-cell heterogeneity in T2D and obesity. The abundance of T2D-related factors and functional β-cell genes in our analysis validates applying DEGAS to islet data to identify disease-associated phenotypes and increase confidence in the novel candidate.”

This part was found at the end of the Background section. We have removed the second sentence to temper the language.